# P53 Deficiency Accelerates Esophageal Epithelium Intestinal Metaplasia Malignancy

**DOI:** 10.3390/biomedicines11030882

**Published:** 2023-03-13

**Authors:** Quanpeng Qiu, Gang Guo, Xiaolong Guo, Xiake Hu, Tianyu Yu, Gaixia Liu, Haowei Zhang, Yinnan Chen, Junjun She

**Affiliations:** 1Department of General Surgery, The First Affiliated Hospital of Xi’an Jiaotong University, Xi’an 710061, China; 2Center for Gut Microbiome Research, Med-X Institute, The First Affiliated Hospital of Xi’an Jiao Tong University, Xi’an 710061, China; 3Department of High Talent, The First Affiliated Hospital of Xi’an Jiaotong University, Xi’an 710061, China

**Keywords:** Barrett’s esophagus, *P53*, acidic bile salts

## Abstract

Barrett’s esophagus (BE) is a precancerous lesion of esophageal adenocarcinoma (EAC). It is a pathological change in which the squamous epithelium distal esophagus is replaced by columnar epithelium. Loss of *P53* is involved in the development of BE and is taken as a risk factor for the progression. We established a HET1A cell line with *P53* stably knockdown by adenovirus vector infection, followed by 30 days of successive acidic bile salt treatment. MTT, transwell assay, and wound closure assay were applied to assess cell proliferation and migration ability. The expression of key factors was analyzed by RT-qPCR, western blotting and immunohistochemical staining. Our data show that the protein expression level of *P53* reduced after exposure to acidic bile salt treatment, and the *P53* deficiency favors the survival of esophageal epithelial cells to accommodate the stimulation of acidic bile salts. Furthermore, exposure to acidic bile salt decreases cell adhesions by repressing the *JAK/STAT* signaling pathway and activating *VEGFR/AKT* in *P53*-deficient esophageal cells. In EAC clinical samples, *P53* protein expression is positively correlated with that of *ICAM1* and *STAT3* and negatively correlated with *VEGFR* protein expression levels. These findings elucidate the role of *P53* in the formation of BE, explain the mechanism of *P53* deficiency as a higher risk of progression for BE formation, and provide potential therapeutic targets for EAC.

## 1. Introduction

Esophageal adenocarcinoma (EAC) is the primary subtype of esophageal cancer. The incidence of EAC has increased year by year in European and American countries [1]. Patients with EAC have poor prognoses. With a 5-year survival rate of less than 20% [2,3], it is one of the leading public health problems that endanger the health of residents. Barrett’s esophagus (BE) is a precancerous lesion of EAC. BE is a pathological change in which the squamous epithelium distal to the esophagus is replaced by columnar epithelium [4]. It is usually an adaptive response caused by prolonged reflux of gastric contents into the esophagus [5,6].

The metaplastic columnar epithelium can be further classified into various types depending on specific histological features. The replacement of esophageal stratified squamous epithelium with intestinal epithelium is known as columnar metaplasia, also referred to as intestinal metaplasia (IM). Barrett’s epithelium appears to progress sequentially from IM to low-grade dysplasia (LGD), high-grade dysplasia (HGD), and finally to invasive EAC [4]. The mechanism for the progression of BE to EAC is not precise. However, three main models are proposed: the ‘gradual accumulation’ model, the ‘born bad’ model, and the ‘catastrophic’ model [7]. In the ‘gradual accumulation’ model, the gene mutations in BE patients gradually increase over time and then slowly progress from nondysplastic BE (NDBE) to LGD and HGD, eventually leading to precancerous lesions [8,9]. The ‘born bad’ model proposes that the degree of malignancy of BE is determined by the patient’s genome if the BE patient carries specific cancerous genes. Patients are also at high risk of becoming cancerous, even with the lowest degree of malignancy NDBE [7]. The ‘catastrophic’ model shows that in patients with non-progressive BE, histological changes due to the occurrence of certain catastrophic events increase the risk of cancer [7,10,11].

The risk of EAC and death in patients with BE is much higher than in normal populations [12]. Currently, the preventive measures for EAC are mainly endoscopy and pathological biopsy to identify patients with BE, carry out early treatment, and reduce the risk of death for EAC [4]. Nevertheless, there is a lack of valid biomarkers to identify high-risk patients among the population of patients with nondysplastic BE, which entirely relies on dysplasia as a sign of increased cancer risk. The presence of dysplasia based on pathologic assessment of BE biopsies is used for patients’ ablation/endoscopic mucosal resection. Polygenic risk scores have also been used for EAC risk assessments, but the results have been less than satisfactory [13].

Tumor suppressor *P53* is the gene with the highest mutation frequency. It is involved in a series of biological processes, including tumor occurrence and development, metastasis and invasion, DNA damage repair, cell cycle, etc. [14,15]. *P53* plays a vital role and is also a key risk factor in the progression of BE. Studies have shown that BE patients with *P53* mutations have a higher risk of cancer than those with *P53* wild type [16]. Thus, loss of *P53* is involved in the progression of BE and plays as a predictor of EAC prognosis [17,18,19]. *P53* mutations are common in patients with BE and are much more frequent in progressive than non-progressive [20,21,22]. Redston et al. found in a prospective study of 1438 patients with BE that *P53* mutation status correlated with the risk of tumor progression, with a worse prognosis for patients with *P53* mutations [23]. However, the exact mechanism of *P53*’s role in the BE progression, especially with exposure to bile acids, remains unclear.

In this study, we simulated the gastroesophageal reflux process by acidic bile salt treatment to explore the effect of *P53* in BE transformation. We found that exposure to acidic bile salt (ABS)/bile salt (BS) reduced the expression of *P53*, and *P53* deficiency promoted cell proliferation. This may be attributed to the adaptation mechanism of the normal esophageal epithelium to acidic bile salt.

## 2. Materials and Methods

### 2.1. Cell Culture

Human esophagus epithelial cells (HET1A) were purchased from American Tissue Culture Collection (ATCC, Bethesda, MD, USA), while OE19 and OE33 were purchased from Beina Chuanglian Biotechnology (Beijing, China) Institute. All the cells were authenticated through Short Tandem Repeat (STR) analysis before being used for experimental studies. The cells were cultured at 37 °C in DMEM medium (HET1A) or 1640 medium (OE19, OE33) supplemented with 10% fetal bovine serum (Gibco; Thermo Fisher Scientific, Waltham, MA, USA) and 1% penicillin-streptomycin (P/S) in a humidified incubator containing 5% CO_2_. To establish an in vitro model for gastroesophageal reflux disease (GERD), HET1A cells were exposed to the bile salt medium once per day for 10 min for up to 30 days.

### 2.2. Collection of EAC Samples

Nine cases of formalin-fixed, paraffin-embedded EAC tissue were collected from the Department of Pathology of the First Affiliated Hospital of Xi’an Jiaotong University (Xi’an, China), and the diagnosis of EAC was jointly determined by two professors of pathology. Hematoxylin and eosin (HE) staining was used to quantify the content of tumor cells in tissues.

### 2.3. Acid and Bile Salt Exposure

HET1A cells were treated with the following experimental media: neutral medium (NC, pH 7.2), which served as control. Neutral bile salt medium (BS, pH 7.2) and acidic bile salt medium (ABS, pH 4.0). Then discard the above medium and wash with PBS three times. Eventually, the cells were cultured in a complete culture medium. Bile salt medium consisted of the following components: glycocholic acid, glycochenodeoxycholic acid, glycodeoxycholic acid, taurocholic acid, taurochenodeoxycholic acid, and taurodeoxycholic acid (TargetMol, Boston, MA, USA) in a 20:15:6:3:3:1 molar concentration ratio, the total concentration was 400 μmol/L [24].

### 2.4. Stable shRNA Transfection

The recombinant plasmid was constructed by inserting shRNA into the plasmid vector *pLKO.1*. The shRNA sequences were as follows: sh*P53*-SiR+: 5′-ccggCCCGGACGATATTGAACAAggatccTTGTTCAATATCGTCCGGGtttttg-3′; sh*P53*-SiR–: 5′-aattcaaaaaCCCGGACGATATTGAACAAggatccTTGTTCAATATCGTCCGGG-3′. The recombinant plasmid vector pLKO.1 and helper plasmid vector pVSVG, pREV, pGAG with lipofectamine 3000 (Invitrogen, Carlsbad, CA, USA) were transfected into 293T cells. The virus was collected after 48 and 72 h. After HET1A cells were infected with the condensed virus for 48 h, puromycin was used to screen the cells.

### 2.5. Western Blot Assay

The cells were lysed by RIPA Lysis Buffer (Beyotime, Shanghai, China) with proteinase and phosphatase inhibitors. Enhanced BCA Protein Assay Kit (Beyotime) was applied to detect the protein concentration. The cell lysate was denatured with 5X SDS buffer and denatured at 95 °C for 5 min. An equal amount of protein to the SDS-PAEG followed by transfer to PVDF Blotting Membrane (Amersham^TM^ Hybond^TM^, Amsterdam, The Netherlands). Next, the PVDF Blotting Membrane was blocked with 5% fat-free dry milk at room temperature for 2 h and incubated with the primary antibodies against *CDX2* (1:1000; Proteintech; Cat. No. 60243-1-Ig), *CK8* (1:1000; Proteintech; Cat. No. 17514-1-AP), *CK13* (1:1000; Proteintech; Cat. No. 10164-2-AP), *ICAM1* (1:1000; Proteintech; Cat. No. 10831-1AP), *OCLN* (1:1000; ABCAM; Cat. No. ab31721), *CLDN2* (1:1000; ABCAM; Cat. No. ab53032), *P53* (1:1000; Proteintech; Cat. No. 60283-2-Ig), *P21* (1:1000; Proteintech; Cat. No. 10355-1-AP), *JAK1* (1:1000; Proteintech; Cat. No. 66466-1-Ig), *STAT3* (1:1000; Proteintech; Cat. No. 10253-2-AP), *VEGFR* (1:1000; Proteintech; Cat. No. 13687-1-AP), *P-AKT* (1:1000; Proteintech; Cat. No. 28731-1-AP), *AKT* (1:1000; Proteintech; Cat. No. 10176-2-AP), *GAPDH* (1:10,000; Proteintech; Cat. No. 60004-1-Ig), *α-Tubulin* (1:10,000; Proteintech; Cat. No. 66031-1-Ig) at 4 °C for overnight. Then the membrane was washed with 1X TBST three times and incubated with corresponding secondary goat-anti-rabbit IgG (1:10,000; Proteintech; Cat. No.) or goat anti-mouse IgG antibodies (1:10,000; Proteintech; Cat. No.) at room temperature for two hours. After the rewash, the protein bands were detected by Chemiluminescent HRP substrate (EMD Millipore, Boston, MA, USA). Protein expression was semi-quantified using ImageJ version 1.46 software (National Institutes of Health, Bethesda, MD, USA).

### 2.6. Immunohistochemistry (IHC) Staining

Five micrometers thick formalin-fixed paraffin-embedded GCA tissue sections were first dewaxed with xylene for 15 min 3 times, then dehydrated by 100% alcohol, 85% alcohol, and 75% alcohol for 5 min, followed by distilled water for 5 min. The tissue was placed in citrate buffer (pH 6.0) for epitope repair. Tissue sections were rinsed in PBS buffer. After blocking with 3% bovine serum albumin (BSA) for 30 min at room temperature, incubate the tissue with the primary antibody overnight at 4 °C. Washed three times with PBS buffer for 15 min each. Incubated the secondary antibody for 1 h at room temperature. After secondary antibody incubation, washed 3 times with PBS buffer for 15 min each on a shaker. Finally, stained the tissue with Harris hematoxylin for 3 min.

### 2.7. Immunofluorescence (IF) Staining

HET1A cells were immobilized with 4% formaldehyde for 15 min and blocked with the mixture of PBS, 1% BSA, 0.1% Triton X-100 for two hours and incubated with the primary antibodies against *Ki67* (1:200; Proteintech; Cat. No. 27309-1-AP) at 4 °C for overnight. And then incubated CoraLite594-conjugated Goat Anti-Rabbit IgG (1:200; Proteintech; Cat. No. SA00013-4) at room temperature for two hours. Finally, DAPI staining was performed, and the images were collected under a fluorescence microscope. Each experiment was repeated three times.

### 2.8. RNA Extraction, RT-qPCR, and RNA-seq Analysis

Total RNA was extracted from cell lines using Trizol reagent (Solarbio, Beijing, China) according to a standard protocol. The Evo M-MLV RT Mix Kit with gDNA Clean for qPCR (Accurate Biotechnology, Wuhan, China) was applied to convert RNA into cDNA. A quantitative real-time polymerase chain reaction (RT-qPCR) was conducted by using Agilent Aria real-time system. The primer sequences were as follows: *CDX2*, 5′-GCAGCCAAGTGAAAACCAGG′ (forward) and 5′-CTGCGGTTCTGAAACCAGATT-3′ (reverse); *CK8*, 5′-TACATGAACAAGGTAGAGCTGG-3′ (forward) and 5′-CCGGATCTCCTCTTCATATAGC-3′ (reverse); *CK13*, 5′-CTCTACCTGTTCAACTCGGTTT-3′ (forward) and 5′-ACAAGCACCAAAGTCAACAAAG-3′ (reverse); *JAK1*, 5′-ATTGAGAACGAGTGTCTAGGGA-3′ (forward) and 5′-CCTTCAGGTCATGCGTGGAC-3′ (reverse); *STAT3*, 5′-CAGCAGCTTGACACACGGTA-3′ (forward) and 5′-AAACACCAAAGTGGCATGTGA-3′ (reverse). Each experiment was repeated three times.

RNA-seq data passing the fast QC (www.bioinformatics.babraham.ac.uk/projects/fastqc/, accessed on 28 March 2022) quality control was filtered to remove the adaptors and low-quality bases using cutadapt. Then Salmon was used for gene-level quantifications, and the DESeq2 R package for the detection of differentially expressed genes with log2 transformed fold-change > 2 and Benjamini–Hochberg adjusted *p*-values < 0.1. The Metascape was used to perform the enrichment analysis of pathways and biological processes with differentially expressed genes [25,26,27].

### 2.9. Cell Proliferation Assay

The cells were seeded into 96-well plates at a density of 5 × 10^3^ cells/well, 10 μL MTT solution (5 mg/mL) was added after 24, 48, 72, 96, or 120 h and then continued to incubate for four hours. Discarded the culture medium from the wells and added 100 μL DMSO, measured the absorbance of wells at 490 nm. Each experiment was repeated three times.

### 2.10. Cell Adhesion Assay

50 μL matrigel (0.04 μg/μL) per well was added to 96-well plates, incubated for one hour at room temperature, removed the remaining matrigel and washed with PBS. Then, seeded the cells into the 96-well plates at a density of 2 × 10^5^ cells/well, cultured at 37 °C for 48 h in a humidified incubator containing 5% CO_2_. Next, removed the culture medium and washed it with PBS, added 10 μL/well MTT solution (5 mg/mL), and continued to incubate for 4 h. Discarded the culture medium from the wells and added 100 μL DMSO, measured the absorbance of wells at 490 nm. Each experiment was repeated three times.

### 2.11. Transwell Migration Assay

HET1A cells (1 × 10^5^ cells) were seeded into transwell inserts with a polyethylene terephthalate membrane with 8μm pore size (Thermo Fisher Scientific, Boston, MA, USA) in 24-well plates with the various 5% FBS. After 24 h, culture media within the transwell inserts were aspirated carefully. Cells were fixed with 2% paraformaldehyde, permeabilized with 0.01% Triton X-100 (Sigma-Aldrich, St. Louis, MO, USA) and stained with crystal violet (Sigma-Aldrich, St. Louis, MO, USA). Cells that did not migrate across the transwell membrane were removed by gently wiping them with a cotton swab. Each experiment was repeated three times.

### 2.12. Cell Culture Wound Closure Assay

Plated the appropriate number of cells in a 6-well plate for 100% confluence in 24 h. In a sterile environment, a 200 μL pipette tip was used to press firmly against the top of the tissue culture plate and swiftly made a vertical wound down through the cell monolayer. Carefully aspirate the media and cell debris. Wound healing was observed under the microscope at 0, 24, and 48 h. Each experiment was repeated three times.

### 2.13. Statistical Analysis

All data were presented as the means ± standard deviation, and statistical analyses were performed by the GraphPad Prism (version 8, San Diego, CA, USA)using Student’s T-test. For all analyses, *p* < 0.05 was considered statistically significant.

## 3. Results

### 3.1. Decreased P53 Expression Level Is Associated with the Progress of BE and EAC

Analyzing data from the cancer genome atlas (TCGA) and Mark Redston et al.’s study [23], we found that in different risk-stratified patients with NDBE, BE indefinite for dysplasia (BE-IND) and BE-HGD, the *P53* mutation rate was positively correlated with risk stratification (Figure 1A). Meanwhile, the frequency of *P53* mutations in EAC was significantly higher than that in BE by bioinformatics analysis (Figure 1B), suggesting that the presence of *P53* mutations in patients with BE would significantly increase the risk of BE progressing to EAC. In patients with EAC, the frequency of *P53* mutations in patients with progressive EAC was higher than that in non-progressive (Figure 1C), indicating that *P53* mutations increase the risk in patients with EAC.

To investigate the role of ABS impacts on *P53* expression, human esophageal epithelial cells HET1A were exposed to the ABS medium once per day for 10 min for up to 30 days (Appendix A). The results indicated dose-dependent inhibition of proliferation of HET1A exposure to ABS or BS for 72 h. With 600 μmol/L exposure to ABS or BS, esophageal epithelial cells were in a state of stagnation (Appendix A). The expression levels of *Ki67* were significantly reduced after 48 h of exposure to 400 μmol/L bile salts in HET1A cells compared with the control group by immunofluorescence staining, and the effect of ABS on HET1A was more obvious than that of BS (Appendix A). To investigate the long-term role of GERD on intestinal metaplasia, the essential process for BE transformation, HET1A cells were exposed to different concentrations of ABS for 48 h. We found that ABS promoted intestinal metaplasia by upregulating the *CDX2* and *CK8* and decreasing *CK13* expression levels in dose-dependence manners (Appendix A).

Meanwhile, BS-treated HET1A cells showed similar results (Appendix A). To evaluate whether *P53* was reduced in esophageal adenocarcinoma cells, *P53* expression was tested in the presence of ABS (Figure 1D) or BS (Figure 1E). To investigate precisely whether bile acids functionally inhibit *P53*, the downstream effector *P21* expression level was detected by western blot. Consistent with that of *P53*, the expression level of *P21* decreased accordingly. Thus, we hypothesized that the repression of *P53* could be due to the adaptation of the normal esophageal epithelium to ABS under GERD conditions, thereby facilitating the viability of HET1A under ABS or BS exposure.

### 3.2. P53 Deficiency Affects Bile Acid-Mediated Growth Inhibition and Intestinal Metaplasia

To investigate the role of *P53* on esophageal epithelium cells’ exposure to bile acid, we established a HET1A cell line with *P53* stably knockdown (sh*P53*) (Figure 2A). After 30 days of successive ABS or BS treatment, cell viability was significantly reduced for both cells with *P53* knockdown and the control group (Figure 2B). The inhibition of ABS was more dramatic than that of BS. Compared with the control group (shCtrl), there was a significant increase in cell proliferation after the *P53* knockdown. To a certain extent, it compensated for the repression of ABS or BS exposure to esophageal epithelium cells (Figure 2B). Consistently, *Ki67* immunofluorescence staining showed that repression dramatically increased with prolonged ABS or BS treatment (Figure 2C–E). To further explore the role of *P53* in the progression of the bile acid-induced BE, the expression of intestinal metaplasia markers *CDX2*, *CK8*, and *CK13* were evaluated. We confirmed that ABS induced intestinal metaplasia of the HET1A cells, and interestingly the *P53* knockdown accelerated this process (Figure 2F and Appendix A). Similar results were also obtained in BS-induced cells (Figure 2G and Appendix A), while the BS-induced intestinal metaplasia was slower than that of ABS.

### 3.3. Loss of P53 Dictates Genes Expression Profile in Esophageal Epithelium CELLS Exposure to Bile Acid

To investigate the expression changes of genes in HET1A cells after exposure to bile acid, we performed RNA-Sequencing analysis in HET1A cells with *P53* knockdown and the control groups with or without ABS treatment. The samples were sequenced in duplicate. The principal component analysis (PCA) showed that the data were tightly clustered between different groups and that there were significant transcriptome differences (Figure 3A). We performed the differential expression analysis (DEA) with an adjusted *p*-value threshold of 0.1 for each set of raw expression measures. In the control group, a total of 1797 differentially expressed genes were identified before and after ABS treatment, including 1004 (55.9%) upregulated genes and 793 (44.1%) downregulated genes; In the *P53* knockdown group, a total of 1550 differentially expressed genes were identified before and after ABS treatment, including 427 (27.5%) upregulated genes and 1123 (72.5%) downregulated genes (Figure 3B). Heatmap showed differentially expressed genes of the control group (left) and *P53* knockdown group (right) before and after ABS treatment (Figure 3C). To identify the role of *P53* on the gene expression of the esophageal epithelium exposed to ABS stimulation, Venn diagram analysis shows differentially expressed genes between groups (Figure 3D). Part I shows 316 genes were upregulated after exposure to ABS in the *P53* knockdown group, but no significant differences in the control group. Part II shows that 59 genes were upregulated after exposure to ABS after the loss of *P53* while downregulated in the control group.

Meanwhile, Part III indicates that 147 genes were downregulated after exposure to ABS in the *P53* knockdown group and upregulated in the control group. Moreover, 912 genes (Part IV) were downregulated after exposure to ABS in the *P53* silenced group but not in the control group. To further clarify the functional role of differentially expressed genes, we conducted GO pathway enrichment analyses (Figure 3E). 316 differentially expressed genes in part I was significantly enriched in response to the hormone pathway, regulation of the cellular response to stress pathway, and heart development pathway. 59 differentially expressed genes contained in part II were significantly enriched in the PID *P53* downstream pathway, Vitamin D receptor pathway, and negative regulation of *MAPK* cascade pathway. 147 differentially expressed genes contained in part III were enriched considerably in response to the cytokine pathway, regulation of cell adhesion pathway, and response to the virus. 912 differentially expressed genes contained in part IV were enriched considerably in the *herpes simplex virus 1* infection pathway, negative regulation of cell population proliferation pathway, and blood vessel development pathway.

### 3.4. P53 Deficiency Regulates Bile Acid-Induced Esophageal Epithelium Cell Adhesion

Cell adhesion is critical in the process of tumorigenesis, and changes in adhesion molecules will affect the interaction between cells-cells and the cell-extracellular matrix, thus affecting the immigration and invasion of tumors. The RNA-seq data indicated that the regulation of the cell adhesion pathway was significantly enriched in the differentially expressed genes contained in Part III (Figure 4A). The collected EAC samples were divided into *P53* high (4 cases) and *P53* low (5 cases) expression groups according to the *P53* protein levels. We found a positive correlation between the expression of *P53* and *ICAM1* (Figure 4B). We further explored the effect of ABS on esophageal epithelial adhesion and elucidated the role of *P53* in this process. HET1A cells were seeded into matrigel to detect their adhesion capacity after ABS or BS treatment (Figure 4C). Cell adhesion was significantly reduced after *P53* knockdown compared with the control group after exposure to ABS (Figure 4D) or BS (Figure 4E).

Furthermore, we found the expression levels of cell adhesion proteins such as *ICAM1*, *OCLN*, and *CLDN2* were decreased. Compared with normal esophageal epithelial cells with exposure to ABS for 20 days or longer, the *P53* deficient HET1A cells showed a significantly decreased protein expression of *ICAM1*, *OCLN*, and *CLDN2* (Figure 4F). BS treatment also showed consistent results with slightly lower differences between the *P53* knockdown group and the control group (Figure 4G). The above results demonstrated that the deficiency of *P53* affects the esophageal epithelial cell adhesion capability when long-term exposed to bile salts, and the silencing of *P53* reduces the expression of *ICAM1*, *OCLN*, and *CLDN2*, thereby favoring cell transformation.

### 3.5. The JAK/STAT Pathway Is Involved in the Regulation of P53 to Bile Salt-Induced Cell Adhesion

RNA-seq results indicated that the response to the cytokine pathway was significantly enriched in the differentially expressed genes contained in Part III (Appendix A). *JAK/STAT* signaling pathway plays an essential role in mediating cytokine signal transduction. Given the turbulence of *JAK/STAT* signaling networks in the EAC, we tested for potential oncogenic activities of these pathways in esophageal epithelial cells with long-term exposure to bile acids. Both the mRNA and protein expression levels of *JAK1* and *STAT3* increased in HET1A cells exposed to ABS or BS for a long time. However, the *JAK/STAT* signaling pathway was repressed after *P53* knockdown (Figure 5A,B and Appendix A), which indicated the *JAK/STAT* signaling pathway was involved in the regulation of *P53* dependence of bile salt-induced cell transformation.

Meanwhile, IHC staining showed that the expression of *P53* protein and *STAT3* protein was positively correlated (Figure 5C). The protein expression levels of *STAT3* and adhesion-related markers, including *ICAM1*, *OCLN*, and *CLDN2*, were severely repressed after 30 days of ABS or BS treatment compared with the control group (Figure 5D,E). Subsequently, normal esophageal cells in the shCtrl group were treated with *STAT3* inhibitor Stattic, which reversed the upregulation of cell adhesion capacity caused by exposure to ABS or BS (Figure 5F,G).

### 3.6. P53 Suppresses the AKT/VEGFR Signaling Pathway to Influence Tumorigenesis

RNA-seq results indicated that the *P53* downstream pathway was significantly enriched in the differentially expressed genes contained in Part II (Appendix A). Mutations in *P53* have been reported leading to the activation of *AKT/VEGFR* signal cascade in some cancers [28,29]. We found the protein expression level of the *VEGFR* signal pathway suppressed in normal esophageal epithelial cells after long-term exposure to ABS or BS treatment. Nevertheless, the expression of *VEGFR* was rescued, and phosphorylation of *AKT* upregulation in HET1A cells after *P53* knockdown (Figure 6A,B). The results of IHC staining showed that the expression level of *VEGFR* in the *P53*-high group was significantly lower than that in the *P53*-low group (Figure 6C). *AKT/VEGFR* signal pathway plays an essential role in cancer cell proliferation, metastasis, and invasion. The activation due to *P53* deficiency could be a potential therapeutic target for EAC. To test this hypothesis, *P53* mutant EAC cells OE19 and OE33 were treated with the *VEGFR* inhibitor sunitinib, followed by analyzing the expression of *Ki67* with immunofluorescence staining. Sunitinib reduced the expression level of *Ki67* and inhibited cell proliferation (Figure 6D,E). In addition, we performed scratch and transwell assay on EAC cells treated with Sunitinib. The results showed that inhibition of *VEGFR* led to the reduction of migration (Figure 6F) and invasion ability (Figure 6G).

## 4. Discussion

BE is becoming an increasingly important disease due to the rapid rise in the incidence of EAC. The development of BE is thought to involve a multistep sequence from metaplastic columnar epithelium to LGD, eventually progressing to HGD and EAC [4]. BE is diagnosed both endoscopically and pathologically. The primary components of gastroesophageal reflux are a mixture of bile salts and acids. Glycocholic acid, glycochenodeoxycholic acid, glycodeoxycholic acid, taurocholic acid, taurochenodeoxycholic acid, and taurodeoxycholic acid are the primary bile acids in gastroesophageal contents [30,31]. The toxic effect of bile acids on cells has been tested in some studies [32,33]. The cell model in this study has limitations. Low pH plays an important role in the transformation of BE [34]. To validate the effect of acids in the process of BE formation, ABS treatment was applied as our experimental group, with BS treatment as the control to explore the role of acid in inducing intestinal metaplasia. In addition, our results showed that after ABS treatment, esophageal epithelial cells have a tendency to transform into BE. Unlike neutral bile salts, intestinal metaplasia markers were significantly upregulated after ABS treatment. This explained the role of acid in the process of intestinal metaplasia and validate the reliability of our gastroesophageal reflux mode.

In this study, esophageal epithelial cells were treated using a bile acid salt mixture and a low pH (pH = 4.0) to mimic the gastroesophageal reflux process in vivo. On the one hand, the expression level of *P53* decreased dose-dependent after exposure to ABS, which induced esophageal epithelium intestinal metaplasia. On the other hand, *P53* deficiency accelerated the progress in epithelial cells with bile acid treatment. *P53* is an essential tumor suppressor factor involved in a range of physiological processes. Thus, we hypothesize that the reduced expression level of *P53* could be the adaptation of normal esophageal epithelium to ABS, thereby facilitating cell survival under the toxic effects of bile salts. The higher positive rate of *Ki67* immunofluorescence staining and cell viability assays support our hypothesis.

In this study, we aimed to explore the mechanism of *P53* in BE transformation. Our results indicated that long-term exposure to bile acid esophagus epithelial cells with *P53* knockdown cell adhesion was reduced due to repression of the *JAK/STAT* signaling pathway (Figure 7). The *JAK/STAT* signaling cascade is a canonical pathway for cytokine signal transduction, including hormones, interleukins, growth factors, etc., involved in the occurrence and development of various diseases, including many types of cancers [35,36,37]. We observed that the treatment of bile salts stimulates the cell adhesion capacity of esophageal epithelial cells. However, the cell adhesion capacity was reduced in *P53* knockdown HET1A cells. Our data suggested that the *JAK/STAT* signaling pathway was involved in this process, and *P53* deficiency reversed the upregulation of the bile acid-induced *JAK/STAT* signaling pathway. For validation, we treated HET1A cells with the *STAT* inhibitor Stattic. The adhesion capacity of the cells was decreased after treatment, compensated for by the increased cell adhesion capacity by the bile salts exposure. In summary, *P53* is critical in the *JAK/STAT* signaling cascade, which is involved in bile acid-induced cell adhesion.

Vascular endothelial growth factor (*VEGF*) can stimulate the proliferation of endothelial cells by interacting with *VEGFR*, which plays a vital role in tumor proliferation and metastasis [38,39]. *P53* mutants showed activation of *AKT/VEGFR* signal cascade [28,40]. Our study showed that prolonged exposure to bile acid treatment downregulated the phosphorylation of *AKT* and expression of *VEGFR* in *P53*-proficient esophagus epithelial cells (Figure 7). However, the *AKT/VEGFR* was activated in *P53* knockdown HET1A cells. The above results suggested that *VEGFR* may be a potential therapeutic target for *P53* mutant EAC. To test this, we treated *P53* mutant EAC cell lines with the *VEGFR* inhibitor Sunitinib. The result demonstrated that the proliferation, migration, and invasion ability of the cells was repressed with a clinical *VEGFR* inhibitor. In short, our results suggested that *P53* could regulate *AKT/VEGFR* signal cascade when exposed to bile acid and further affects EAC cell proliferation, migration, and invasion.

## 5. Conclusions

In sum, this study explored the mechanism of *P53* in BE transformation of esophagus epithelial cells. Our results show that during BE formation, the deficiency of *P53* facilitates cells to adapt to the toxic effect of bile salts by upregulating the *AKT/VEGFR* signal cascade, in the meantime, promotes the intestinal epithelial metaplasia process. Our findings improve the understanding of the specific molecular mechanisms of BE transformation and provide novel potential targets for EAC with *P53* mutations.

## Figures and Tables

**Figure 1 biomedicines-11-00882-f001:**
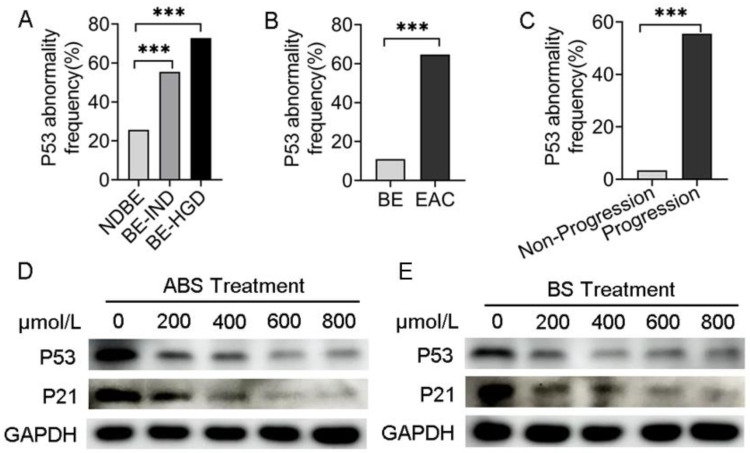
*P53* deficiency is associated with the progression of BE and EAC. (**A**) *P53* abnormal frequency in NDBE, BE-IND and BE-HGD. (**B**) *P53* abnormal frequency in BE and EAC. (**C**) *P53* abnormal frequency in progressive and non-progressive EAC. (**D**) P53 and *P21* protein expression after treatment with ABS titration in HET1A cells after 72 h. (**E**) *P53* and *P21* protein expression after treatment with BS titration in HET1A cells after 72 h. BE: Barrett’s esophagus; EAC: esophageal adenocarcinoma; NDBE: nondysplastic BE; BE-IND: BE indefinite for dysplasia; BE-HGD: BE with high-grade dysplasia; ABS: acidic bile salt medium; BS: neutral bile salt medium. Data were statistically analyzed using the student *t* test. *** *p* < 0.001.

**Figure 2 biomedicines-11-00882-f002:**
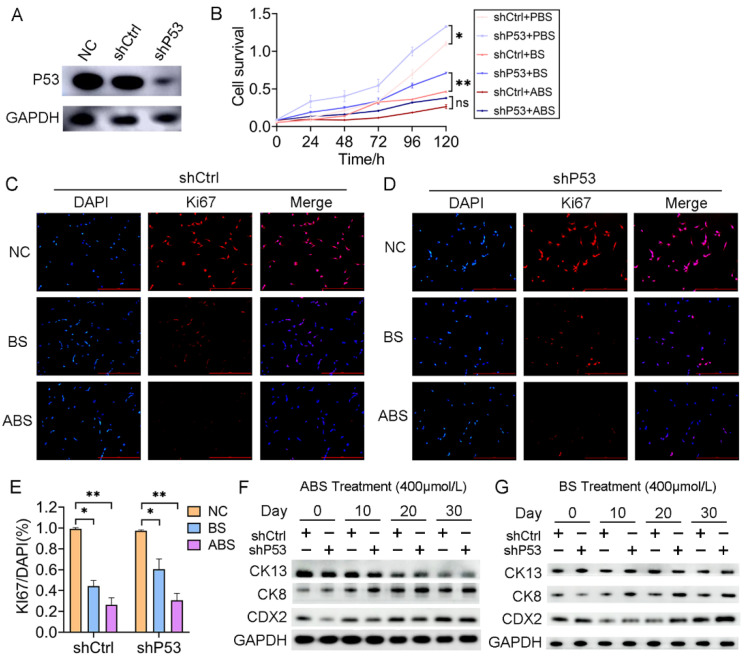
*P53* deficiency affects bile acid-mediated cell proliferation repression and intestinal metaplasia. (**A**) HET1A cells with *P53* knockdown by shRNA, Western Blot assay of *P53* expression level. (**B**) The proliferation of HET1A cells with or without *P53* knockdown after 30 days of continuous exposure to ABS or BS. (**C**) Representative immunofluorescence of *Ki67* images of HET1A cells with 400 μmol/L ABS or BS treatment. (**D**) Representative immunofluorescence of *Ki67* images of *P53* knockdown HET1A cells with 400 μmol/L ABS or BS treatment. (**E**) Quantification of *Ki67* expression levels in HET1A cells with or without *P53* knockdown stimulated by ABS and BS. (**F**) After continuous exposure to ABS for denoted time, the expression levels of squamous epithelium marker *CK13*, columnar epithelium marker *CK8* and intestinal metaplasia marker *CDX2*. (**G**) After continuous exposure to BS for denoted time, the expression levels of *CK13*, *CK8*, and *CDX2*. +: Treated accordingly; −: Not treated accordingly. Data were statistically analyzed using the student *t*-test. * *p* < 0.05, ** *p* < 0.01, ns: none significance.

**Figure 3 biomedicines-11-00882-f003:**
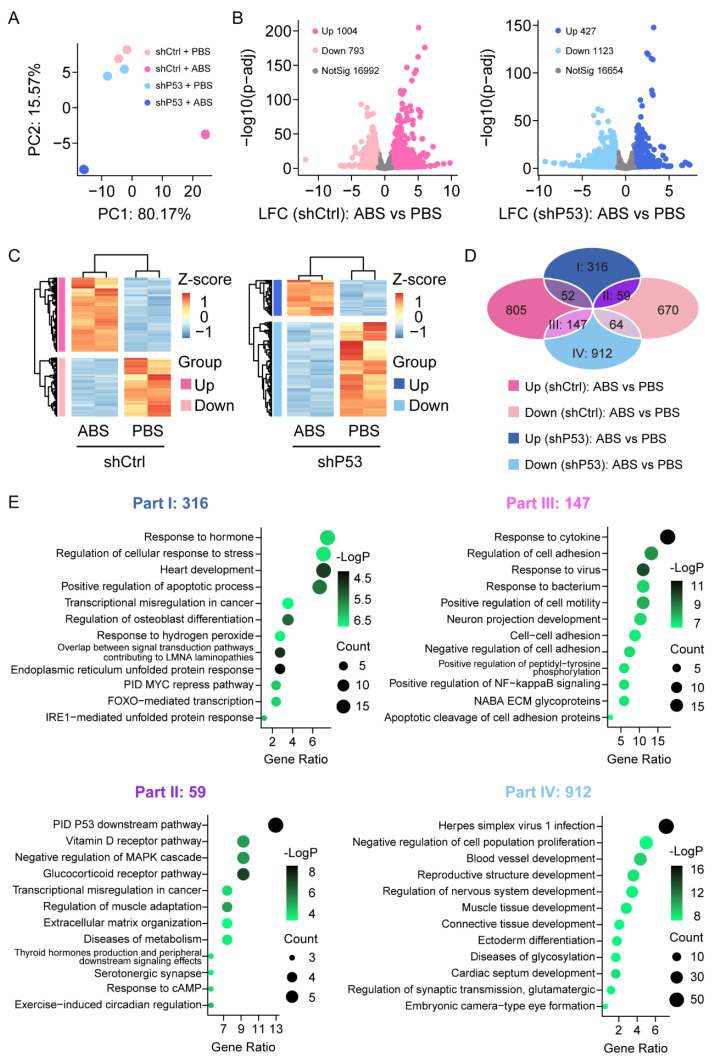
Transcriptome of the esophageal epithelium cells with or without *P53* knockdown after ABS treatment. (**A**) PCA of transcriptome data for HET1A with or without *P53* knockdown after 30 days of exposure to ABS. (**B**) Volcanic graph analysis of gene expression of the control group (left) and *P53* knockdown HET1A cells (right). (**C**) Heatmap of genes expression of the control group (left) and *P53* knockdown group (right) before and after exposure to ABS. (**D**) Venn diagram analysis of gene expression various in *P53* silenced HET1A cells and control group with or without ABS treatment. (**E**) GO enrichment analysis of differentially expressed genes in Parts I–IV denoted parts in (**D**).

**Figure 4 biomedicines-11-00882-f004:**
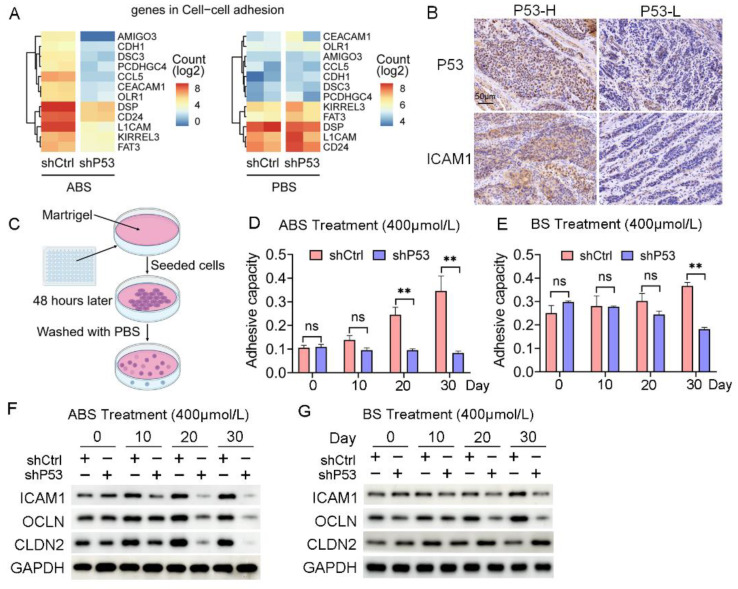
*P53* dependence of bile acid-induced cell adhesion. (**A**) Heatmap of gene expression in cell adhesion pathway after exposure to ABS (left) and PBS (right) of *P53* knockdown and control group. (**B**) Representative images of *P53* and *ICAM1* IHC staining. (**C**) Schematic outline of cell adhesion assay. (**D**) Quantification of cell adhesion capacity of HET1A cell with or without *P53* knockdown after ABS treatment. (**E**) Quantification of cell adhesion capacity of HET1A cell with or without *P53* knockdown after BS treatment. (**F**) The expression levels of cell adhesion markers *ICAM*1, *OCLN* and *CLDN2* after continuous exposure to ABS. (**G**) The expression levels of cell adhesion markers *ICAM1*, *OCLN* and *CLDN2* after continuous exposure to BS. +: Treated accordingly; −: Not treated accordingly. Data were statistically analyzed using the student *t*-test. ** *p* < 0.01, ns: none significance.

**Figure 5 biomedicines-11-00882-f005:**
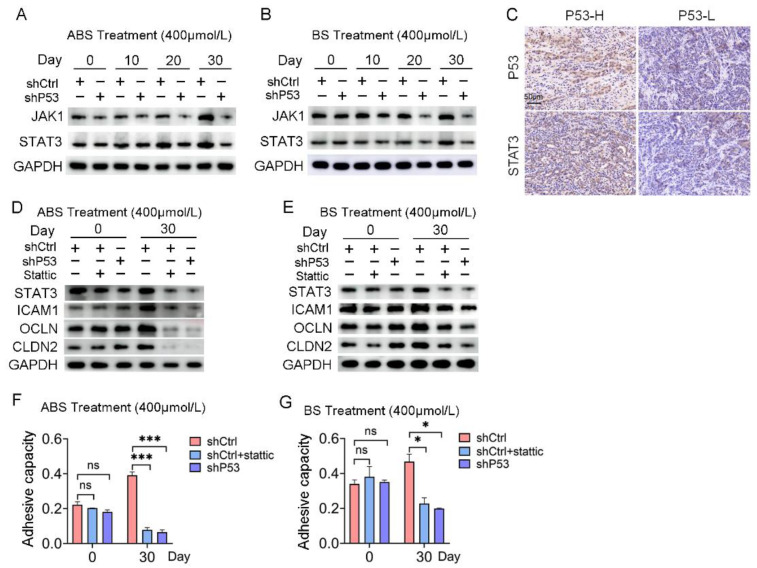
The *JAK/STAT* signaling pathway is involved in *P53* dependence on bile acid-induced cell adhesion. (**A**) The expression levels of *JAK1* and *STAT3* after ABS treatment. (**B**) The expression levels of *JAK1* and *STAT3* after BS treatment. (**C**) Representative images of *P53* and *STAT3* IHC staining. (**D**) Expression of cell adhesion markers *ICAM1*, *OCLN*, *CLDN2* and *STAT3* in HET1A cells with *P53* knockdown or Stattic treatment after ABS stimulation for 30 days. (**E**) Expression of cell adhesion markers *ICAM1*, *OCLN*, *CLDN2* and *STAT3* in HET1A cells with *P53* knockout or Stattic treatment after BS stimulation for 30 days. (**F**) Cell adhesion capacity after exposure to ABS for 30 days after STAT3 inhibitor Stattic treatment. (**G**) Cell adhesion capacity after exposure to BS for 30 days after Stattic treatment. +: Treated accordingly; −: Not treated accordingly. Data were statistically analyzed using the student *t*-test. * *p* < 0.05, *** *p* < 0.001, ns: none significance.

**Figure 6 biomedicines-11-00882-f006:**
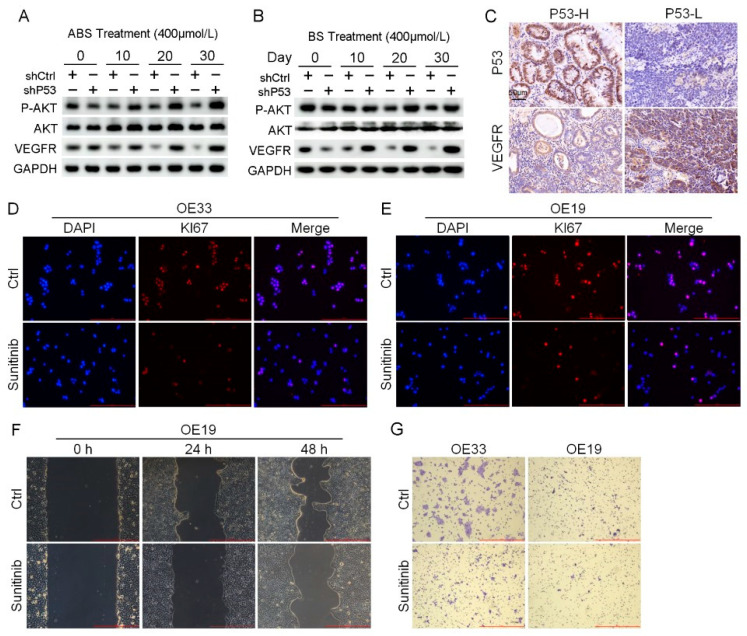
Venerability of *VEGFR* signaling pathway in *P53* deficient EAC cells. (**A**) The protein expression level of *AKT*, *VEGFR* and phosphorylation of *AKT*. (**B**) The protein expression level of *AKT*, *VEGFR* and phosphorylation of *AKT*. (**C**) Representative images of *P53* and *VEGFR* IHC staining. (**D**,**E**) Representative images of OE33 cells (**D**) and OE19 cells (**E**) by *Ki67* immunofluorescence after treatment with 2 μmol/L Sunitinib. (**F**) OE33 cell migration capacity after treatment with 2 μmol/L Sunitinib. (**G**) Transwell assay for OE33 and OE19 cell lines after treatment with 2 μmol/L Sunitinib. +: Treated accordingly; −: Not treated accordingly. Data were statistically analyzed using the student *t*-test.

**Figure 7 biomedicines-11-00882-f007:**
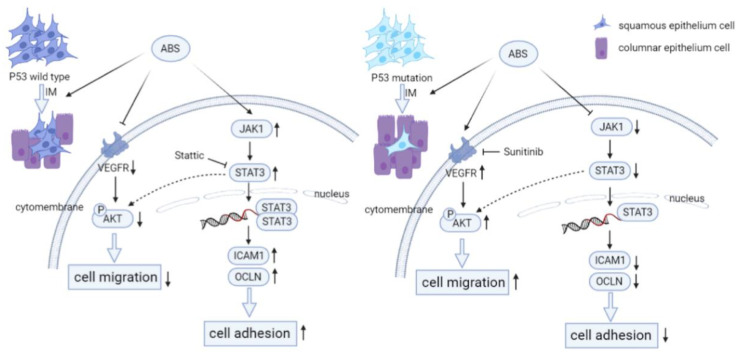
*P53* deficiency promotes esophageal epithelium intestinal metaplasia malignancy after exposure to Acid Bile Salt. (**Left**) In *P53* wild-type esophageal epithelial cells, ABS stimulation induces intestinal metaplasia of the esophageal epithelium and reduce the invasive ability of HET1A cells by downregulating the expression level of *VEGFR* and inhibiting the phosphorylation modification of *AKT*. Meanwhile, the *JAT/STAT* pathway is activated, and the downstream cells’ adhesion-related proteins (*ICAM1, OCLN, CLDN2*) are highly expressed, resulting in the enhancement of cell adhesion ability. (**Right**) Compared with wild-type esophageal epithelial cells, *P53* deficiency promotes ABS-induced intestinal metaplasia of the esophageal epithelium. And the expression level of *VEGFR* and the phosphorylation of *AKT* are upregulated to enhance the invasive ability of HET1A cells. However, *JAK/STAT* pathway is inhibited, and the downstream cell adhesion-related proteins are repressed, reducing cell adhesion ability.

## Data Availability

The data presented in this study are available upon request from the corresponding author.

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
