# Peer review of "P53 Deficiency Accelerates Esophageal Epithelium Intestinal Metaplasia Malignancy"

_biomedicines, 2023, doi:10.3390/biomedicines11030882_

Round 1

Reviewer 1 Report

In the present basic science article Qius et al evaluated the effects of bile acid exposure on p53-related carcinogenesis in a cell culture model of Barrett esophagus (BE). Main comments:

1) The title is hard to read and should be re-written.

2) Page 2 lines 77-78: this sentence is false, as intestinal metaplasia (IM) is induced by gastric acid (HCl), not by bile salts.

3) A paragraph in Methods section describing statistical analysis is lacking.

4) The main problem of this paper is that the model is not fit for investigating BE. Indeed Authors simply used esophageal cultured cells and exposed them to acid bile salts (ABS), therefore they did not demonstrate if IM really took please. Moreover, as mentioned before, IM is induced by HCl, not by ABS. therefore the whole study design is wrong.

Author Response

In the present basic science article Qius et al evaluated the effects of bile acid exposure on p53-related carcinogenesis in a cell culture model of Barrett esophagus (BE). Main comments:

  1. The title is hard to read and should be re-written.

Response 1

We appreciate your comments and suggestions. We rewrote the title of the article to make it more concise and understandable.

  1. Page 2 lines 77-78: this sentence is false, as intestinal metaplasia (IM) is induced by gastric acid (HCl), not by bile salts.

Response 2

We apologize for the unclear description, and we have corrected this part of the article. (Page 2 lines 76-77)

  1. A paragraph in Methods section describing statistical analysis is lacking.

Response 3

We added the missing methods section describing statistical analysis. (Page4 lines 202-205)

  1. The main problem of this paper is that the model is not fit for investigating BE. Indeed, authors simply used esophageal cultured cells and exposed them to acid bile salts (ABS), therefore they did not demonstrate if IM really took please. Moreover, as mentioned before, IM is induced by HCl, not by ABS. therefore the whole study design is wrong.

Response 4

Thank you for your questions, cell model of this study is indeed a limitation. In this study, we used ABS to mimic the gastroesophageal reflux process in vitro, reflux contents consist mainly of acid and bile salts[Ref. 31].In Huo et al.’s and Zhang et al.’s study, cells were also exposed to a mixture of acids and bile salts for a long time to mimic the clinical process of gastroesophageal reflux[Ref. 32, DOI:10.1053/j.gastro.2018.09.046]. Therefore, we used acid bile salts treatment to construct an in vitro gastroesophageal reflux model. We chose ABS treatment as our experimental group, and used BS treatment as a control to explore the role of acid in inducing intestinal metaplasia. In addition, our results showed that after ABS treatment, esophageal epithelial cells have a tendency to transform into BE. Compared with neutral bile salts, intestinal metaplasia markers were more significantly upregulated after ABS treatment, which explained the role of acid in the process of intestinal metaplasia, and proved that the construction of our gastroesophageal reflux model was feasible.

Reviewer 2 Report

This is a well designed , executed and written experimental article. There are however far too many graphs and illustrations. I suggest to reduce them by 50% 

Author Response

This is a well designed , executed and written experimental article. There are however far too many graphs and illustrations. I suggest to reduce them by 50%.

Response:

Thank you for your suggestions. We have reduced graphs and illustrations.

Reviewer 3 Report

The authors simulated the gastroesophageal reflux process to explore the effect of P53 on the intestinal metaplasia process induced by bile acid. They found that exposure to acidic bile salt/bile salt reduced the expression of P53, and P53 deficiency promoted cell proliferation and intestinal metaplasia. Furthermore, exposure to bile acids decreases cell adhesions by expression the JAK/STAT signaling pathway and activation VEGFR/AKT in P53 with that of ICAM1 and STAT3, and negatively correlated with VEGFR protein expression levels. They suggested that P53 could regulate AKT/VEGFR signal cascade when exposed to bile acid for intestinal metaplasia, and further affects EAC cell proliferation, migration, and invasion. This manuscript is worth being published in this Biomedicines and gives us instructive message. However, there were numerous typing errors including the mixture of half-size and full-size font characters.

Author Response

The authors simulated the gastroesophageal reflux process to explore the effect of P53 on the intestinal metaplasia process induced by bile acid. They found that exposure to acidic bile salt/bile salt reduced the expression of P53, and P53 deficiency promoted cell proliferation and intestinal metaplasia. Furthermore, exposure to bile acids decreases cell adhesions by expression the JAK/STAT signaling pathway and activation VEGFR/AKT in P53 with that of ICAM1 and STAT3, and negatively correlated with VEGFR protein expression levels. They suggested that P53 could regulate AKT/VEGFR signal cascade when exposed to bile acid for intestinal metaplasia, and further affects EAC cell proliferation, migration, and invasion. This manuscript is worth being published in this Biomedicines and gives us instructive message. However, there were numerous typing errors including the mixture of half-size and full-size font characters.

Response:

We appreciate your comments and suggestions. We checked the spelling throughout the manuscript and corrected as marked in the text.

Round 2

Reviewer 1 Report

Answer to point 4 is not satisfactory